# Full-Perception Robotic Surgery Environment with Anti-Occlusion Global–Local Joint Positioning

**DOI:** 10.3390/s23208637

**Published:** 2023-10-22

**Authors:** Hongpeng Wang, Tianzuo Liu, Jianren Chen, Chongshan Fan, Yanding Qin, Jianda Han

**Affiliations:** 1College of Artificial Intelligence, Nankai University, Tianiin 300353, China; hpwang@nankai.edu.cn (H.W.); liutz@mail.nankai.edu.cn (T.L.); chenjr@mail.nankai.edu.cn (J.C.); csfan@mail.nankai.edu.cn (C.F.); 2Institute of Intelligence Technology and Robotic Systems, Shenzhen Research Institute of Nankai University, Shenzhen 518083, China; 3Engineering Research Center of Trusted Behavior Intelligence, Ministry of Education, Nankai University, Tianjin 300350, China

**Keywords:** robotic surgery, global–local joint positioning, data fusion, occlusion avoidance

## Abstract

The robotic surgery environment represents a typical scenario of human–robot cooperation. In such a scenario, individuals, robots, and medical devices move relative to each other, leading to unforeseen mutual occlusion. Traditional methods use binocular OTS to focus on the local surgical site, without considering the integrity of the scene, and the work space is also restricted. To address this challenge, we propose the concept of a fully perception robotic surgery environment and build a global–local joint positioning framework. Furthermore, based on data characteristics, an improved Kalman filter method is proposed to improve positioning accuracy. Finally, drawing from the view margin model, we design a method to evaluate positioning accuracy in a dynamic occlusion environment. The experimental results demonstrate that our method yields better positioning results than classical filtering methods.

## 1. Introduction

With the development of robotics science, robots have begun to assist or participate in the process of surgery. There are already commercial robotic systems for different types of surgery, such as the Da Vinci surgical system (Intuitive Surgical, Inc., Sunnyvale, CA, USA) and VELYS surgical robots (DePuy Synthes, Inc., Warsaw, IN, USA). Whether it is a master–slave robot operated by doctors or a robot with the ability to perform surgery autonomously, it is necessary to accurately track the position and orientation of the end-effector of the robotic arm during the operation to ensure the safety of the surgical process. In existing robotic surgery, optical tracking systems are usually used to obtain the exact position and orientation of the end-effector.

In image-guided surgery (IGS) employing optical tracking systems, hand–eye calibration is conducted prior to commencing the operation to determine the transformation matrix between the coordinate system of the operating arm and the coordinate system of the camera system. Throughout the operation, it is imperative to maintain an unchanging relative position between these systems. Additionally, due to the physical nature of optical tracking systems, it is essential to prevent any occlusion between the cameras and the target; otherwise, it may lead to an interruption of the surgical process. Occlusion has consistently been a central challenge in optical positioning technology. Consequently, in the existing surgical environment, intricate spatial constraints exist among humans and robots as well as between different robots. Due to local positioning space limitations and for the sake of surgical process safety, the current robotic surgery tends to maintain relative isolation between humans and robots, as well as among different robots. To further enhance the intelligence of the surgical setting, it becomes crucial to comprehensively perceive the presence of both people and robots in the operating room. Hence, the concept of a full-perception robotic surgery environment is introduced. Currently, there is limited existing research in this area.

This paper constructs a global–local joint positioning framework by integrating local and global positioning information. It employs global base markers as the bridge between global and local coordinates and utilizes a data fusion method to track the targets. Within the data fusion method, data characteristics are considered to enhance positioning accuracy. In dynamic operational scenarios, an evaluation method for assessing the positioning system within dynamic scenes, based on the view margin model, is introduced to evaluate the framework. Figure 1 depicts a scene from robot-assisted surgery, featuring multiple targets partially obstructed and observed by various positioning devices.

## 2. Related Works

Commonly employed tracking systems in operating rooms encompass mechanical trackers [1], electromagnetic tracking systems [2], ultrasound tracking systems [3], optical tracking systems (OTS) [4,5,6], among others. OTS, distinguished by its high precision and robustness, finds widespread utilization. OTS can be categorized into active tracking and passive tracking, both achieving nearly identical accuracy [7]. Binocular OTS has gained extensive application in fields such as orthopedic surgery [8] and neurosurgery [9]. The inherent characteristics of optical cameras contribute to occlusion as a significant challenge in IGS. A multi-camera system emerges as an intuitive solution [10,11,12]. Researchers [10] established an eight-camera system in an intraoperative electronic radiation therapy scenario, conducting testing via phantom experiments. The positioning error remained below 2 mm; however, no other moving objects were present in the scene. Another research group [13] combined two sets of OTS to expand the system’s field of view. An alternative approach involves proactively preventing occlusion. Researchers [14] incorporated OTS into a robot with four degrees of freedom, enabling it to move when the target is obscured, effectively avoiding occlusion and transforming the issue into a motion planning and optimization problem for the robot. Additionally, another research team [15] designed a sensor fusion method utilizing optical and inertial data for posture estimation. These methods operate under the assumption of an ideal surgical process with no mutual interference between robots, humans, or other robots.

With the further application of robot technology, researchers have proposed [16] that perceiving the overall environment in robotic surgery scenes can help improve surgical performance and patient safety. Researchers [17] combined team motion behavior with physiological signals to analyze the entire surgical process and found that perceiving behavior and video recording in the operating room can help improve surgical and teaching quality. In addition, intraoperative imaging can be used for robot-assisted surgery [18], which requires some equipment to be moved during the procedure. When the perception range expands to the entire operating room, simple local positioning methods are no longer sufficient, and it is necessary to introduce global positioning information.

Optical motion capture (OMC) is an effective method for obtaining stable and accurate global positioning information. The accuracy of commercial motion capture systems can reach the sub-millimeter level. OMC has found utility in medical information acquisition within larger workspaces [19,20]. Researchers [21] have delved into the data characteristics related to tracking body posture in various anatomical regions. Studies [22,23,24] have showcased that multi-camera systems exhibit superior accuracy at the center of their field of view. However, the operating room presents a complex and dynamic occlusion environment. Leveraging a robotic arm, researchers [22] analyzed the correlation between the multi-camera system’s accuracy within a defined workspace and the presence of occluded and mis-calibrated cameras, factors that necessitate consideration in real-world scenarios. In the case of binocular optical tracking systems (OTS), accuracy exhibits a linear relationship with distance within the effective working range [25]. Nevertheless, as a multi-camera system, OMC error manifests complex characteristics [26]. An exploration of error propagation in triangulation [27] aids in determining the impact of camera positioning on 3-D reconstruction accuracy. However, obtaining OMC accuracy in a randomly occluded environment cannot be achieved through static field-of-view calculations. A study [28] introduced a strategy for camera placement in dynamic scenes based on a probabilistic occlusion model, offering a framework to address the aforementioned issues in dynamic settings.

In this paper, we introduce a global–local joint positioning method designed for a full-perception surgery environment. This method seamlessly integrates data from both an optical motion capture system and a binocular optical tracking system, enabling the system to effectively handle occlusion and provide stable and highly accurate global positioning information. Our primary approach involves the creation of a global–local positioning framework to ensure the reliable output of positioning information. Furthermore, we employ an improved filtering algorithm that takes into account the actual characteristics of the data, resulting in enhanced positioning accuracy. The system’s architecture is depicted in Figure 2. To assess the dynamic positioning accuracy, we have developed a testing and evaluation method based on the view margin model, which substantiates the effectiveness of our approach.

The main contributions of our work can be summarized as follows:We innovatively introduced the concept of a full-perception robotic surgery environment and established a global–local joint positioning framework;We enhanced positioning accuracy by integrating the biased Kalman filter algorithm with data characteristics;We devised an evaluation method based on the view margin model for assessing dynamic positioning accuracy, thereby demonstrating the superiority of our method.

## 3. Global-Local Positioning Method

### 3.1. Unified Coordinate Expression

The OTS employed in robot-assisted surgery scenes involves the installation of specific-sized passive reflective spheres on rigid bodies with distinct geometric characteristics. It aims to identify and match corresponding reflective spheres within the unobstructed field of view of binocular optical cameras, enabling the calculation of the spatial position relationship between the target and the camera coordinate system through the parallax of the same target imaged in both binocular cameras. However, in such a system, when obstacles enter the field of view of one or both cameras and obstruct certain reflective points, the binocular optical system fails to recognize the geometric characteristics of this group of reflective points, thereby affecting the tracking task. The conventional solution to this challenge entails manual adjustments to the position and orientation of the optical tracker during surgery, albeit at the expense of surgical process continuity.

As depicted in Figure 3, we attach distinct rigid bodies with specific geometric characteristics to various targets within the surgical environment. These targets encompass the fixed position of the base coordinate system, serving as the global reference coordinate system, as well as the surgical instruments, the base of the robotic arm, and other mobile targets.

Taking the optical navigation robotic arm as an example, the base coordinate system $\left\{Base\right\}$ is denoted as ${}_{Base}^{OMC}T$ within the motion capture system. The binocular optical tracking system is mounted on a robotic arm with four or more degrees of freedom, allowing us to establish the relationship between ${}_{OTS}^{Robot}T$ through Denavit–Hartenberg (D-H) parameters and hand-eye calibration. Typically, the $\left\{Robot\right\}$ coordinate system can be tracked within the motion capture system and is represented as ${}_{Robot}^{OMC}T$. The expression of the tool coordinate system $\left\{Tool\right\}$ in the binocular optical tracking system is ${}_{Tool}^{OMC}T$. As a result, the representation of the target in the global coordinate system can be expressed as follows:(1)$${}_{Tool}^{Base}T={}_{Base}^{OMC}T^{-1}{}_{Robot}^{OMC}T{}_{OTS}^{Robot}T{}_{Tool}^{OTS}T$$

Specifically, when $\left\{Base\right\}$ is simultaneously visible in the binocular optical tracking system, the formula above can be simplified to:(2)$${}_{Tool}^{Base}T={}_{Base}^{OTS}T^{-1}{}_{Tool}^{OTS}T$$

Based on the equation above, a unified expression for the target in the global coordinate system is obtained.

It is worth noting that in this example, the rotation matrix of the mobile device is ${}_{OTS}^{Robot}T$, which is determined through hand–eye calibration and the D-H parameters of the robotic arm. Therefore, even though these parameters can be updated and calibrated during dynamic processes, as long as the calculations are completed during system initialization, temporary occlusions occurring in subsequent processes will not disrupt the system’s normal operation.

### 3.2. Data Fusion Method

Our method utilizes optical motion capture (OMC) to provide stable global information, but there exists a substantial disparity between the operating room environment and the typical OMC use scenario. Oftentimes, OMC systems are deployed in open-field-of-vision settings where most optical cameras within the system can observe the same target. Consequently, these setups offer high accuracy and yield true values for measurement tasks like attitude tracking.

However, occlusion within the operating room scene is dynamic and intricate. Take the two reflectors shown in Figure 4, which are rigidly linked, as an example. The curve shows the error between the measured value and the true value between the reflecting balls. The maximum error in the figure exceeds 0.5 mm; in some extreme cases, this error can reach 2–3 mm. This demonstrates that changes in available lines of sight can significantly amplify system errors due to insufficient data being incorporated into calculations. Another source of error stems from incomplete calibration, leading to distortions akin to those encountered in camera imaging when determining spatial positions. This characteristic is reflected in our experiments in Section 5. For precise positioning in the operating room environment, these factors must be taken into account.

The Kalman filter (KF) is an optimized autoregressive data processing algorithm used for reconstructing the system’s state vector through measured values. It operates recursively in the logical sequence of “prediction-update” and is primarily employed for solving estimation problems in linear systems. The fundamental aim is to find the estimated value ${\hat{x}}_{k}$ of the state vector $x_{k}\in R^{n}$ at time *k* while minimizing the mean square error. We utilize this approach for data fusion. To address the aforementioned error factors, the biased Kalman filter [29] and compensated state vectors are employed for processing.

The state equation and measurement equation of a linear system can be represented as follows:(3)$$\begin{array}{cc} x_{k} & =F_{k}x_{k-1}+w_{k}\\ y_{k} & =H_{k}x_{k}+v_{k} \end{array}$$

Bias the state estimator $x_{k}$ using the partial parameter matrix $M_{k}$, where:(4)$$M_{k}=diag\left[m_{k}^{i}\right],i=1,2,\ldots ,n$$

Expressed as ${\hat{x}}_{k}^{b}=M_{k}{\hat{x}}_{k}^{}$.

Then, the mean square error of ${\hat{x}}_{k}^{b}$ can be formulated as:(5)$$MSE\left[{\hat{x}}_{k}^{b}\right]=M_{k}Tr\left[P_{k}\right]M_{k}^{T}+x_{k}{\left(M_{k}-I\right)}^{T}\left(M_{k}-I\right)x_{k}^{T}$$

The mean square error of each element ${\hat{x}}_{k}^{bi}$ in ${\hat{x}}_{k}^{b}$ is:(6)$$MSE\left[{\hat{x}}_{k}^{bi}\right]=\mathrm{var}\left[{\hat{x}}_{k}^{bi}\right]+{\left(E\left[{\hat{x}}_{k}^{bi}-x_{k}^{i}\right]\right)}^{2}$$
where ${\hat{x}}_{k}^{bi}={\left(m_{k}^{i}\right)}^{2}{\hat{x}}_{k}^{i}$.

The offset mean square error is a function of the partial parameter $m{}_{k}^{i}$. The optimal outcome can be achieved by taking the derivative of $m{}_{k}^{i}$:(7)$$m_{k}^{i}=\frac{{\left(x_{k}^{i}\right)}^{2}}{\mathrm{var}\left[{\hat{x}}_{k}^{i}\right]+{\left(x_{k}^{i}\right)}^{2}}$$

Substituting Equation (7) into Equation (6):(8)$$MSE\left[{\hat{x}}_{k}^{bi}\right]=\frac{{\left(x_{k}^{i}\right)}^{2}}{\mathrm{var}\left[{\hat{x}}_{k}^{i}\right]+{\left(x_{k}^{i}\right)}^{2}}MSE\left[{\hat{x}}_{k}^{bi}\right]\leq MSE\left[{\hat{x}}_{k}^{i}\right]$$

The biased estimation result outperforms the unbiased result in terms of mean square error.

Building upon this, we increase the dimension of the state vector and introduce a deviation term to estimate the hop error in global positioning. This enhancement enables the algorithm to counter hop errors when only global information is accessible and update the deviation term when local information is introduced. The specific implementation method is as follows:

Define the extended dimension state vector as $x_{k}={\left[\begin{array}{ccccccccc} p_{x} & p_{y} & p_{z} & v_{x} & v_{y} & v_{z} & b_{x} & b_{y} & b_{z} \end{array}\right]}^{T}$.

The global positioning information initializes the system state. When the local positioning system can observe the target, ${}_{OTS}^{Robot}T$ is calculated.

Since the target motion in the operating room scene is stable, a uniform motion model is used locally to predict the target state:(9)$${{\hat{x}}_{k}}^{\prime }=F_{k}{\hat{x}}_{k-1}$$

The covariance matrix is calculated according to the classical Kalman filtering method:(10)$${P_{k}}^{\prime }=F_{k}P_{k-1}F_{k}^{T}+Q_{k}$$

To evaluate the input data, employ distinct observation matrices to represent this variation: (11)$$H_{k}=\left\{\begin{array}{cc} \;\left[\begin{array}{ccc} I_{3} & 0 & I_{3} \end{array}\right], & \mathrm{Global}\;\mathrm{data}\\ \;\;\left[\begin{array}{ccc} I_{3} & 0 & 0 \end{array}\right], & \mathrm{Local}\;\mathrm{data} \end{array}\right.$$

Calculate Kalman gain:(12)$$\begin{array}{c} G_{k}={P_{k}}^{\prime }H_{k}^{T}{\left(H_{k}{P_{k}}^{\prime }H_{k}^{T}+R_{k}\right)}^{-1}\\ P_{k}=\left(I-G_{k}H_{k}\right){P_{k}}^{\prime } \end{array}$$

Unbiased estimation of the state at the next time:(13)$${{\hat{x}}_{k}}^{\prime \prime }={{\hat{x}}_{k}}^{\prime }+G_{k}\left(y_{k}-H_{k}{{\hat{x}}_{k}}^{\prime }\right)$$

Introducing partial parameter matrix $M_{k}$:(14)$$\begin{array}{c} {\hat{x}}_{k}=M_{k}{{\hat{x}}_{k}}^{\prime \prime }\\ P_{k}=M_{k}{P_{k}}^{\prime }M_{k}^{T} \end{array}$$

Carry ${\hat{x}}_{k}$ and $P_{k}$ to the next iteration. This method is referred to as the compensated bias Kalman filter (CBKF). The experimental results will be presented in Section 5.

## 4. Dynamic Positioning Evaluation Method for Full-Perception Surgery Scene

For the proposed joint positioning task in the full-perception surgery environment, occlusion is generated in the surgical scene due to the diversity of environments and tasks, making it both random and dynamic. Consequently, the traditional method of evaluating static positioning accuracy, which focuses on absolute positioning accuracy, falls short in assessing the overall system performance. To address this, a dynamic scene positioning evaluation method based on the view margin model is introduced. This method is designed to comprehensively and scientifically evaluate the positioning effectiveness.

In defining this method, we recognize that different locations in space pose varying levels of difficulty during the positioning process. Therefore, the tasks we design aim to encompass these diverse spatial volumes, allowing the positioning results to reflect the system’s overall performance throughout the space. Given the relative randomness of camera placement and the mobility of local positioning, quantifying the accuracy distribution across different spatial positions is not achieved simply by overlapping visual cones. To address this challenge, we first analyze the static factors influencing positioning accuracy. Then, we integrate these findings with the dynamic process using the view margin model, utilizing a specific metric to assess the difficulty of each region in the measured space. Subsequently, an evaluation task based on these considerations is designed.

### 4.1. Static Factor Analysis

In the optical positioning system, whether it is an optical motion capture (OMC) or a binocular optical tracking system (OTS) composed of infrared optical cameras, the fundamental principle involves 3-D reconstruction through triangulation. It calculates the 3-D coordinates of a target in space by measuring the parallax of the same target in different cameras. Since optical camera imaging relies on the principle of pinhole imaging, resolution plays a crucial role in measurement accuracy.

Typically, standard reflectors with a radius ranging from 5 to 10mm are used. Given specific camera parameters, the visibility of image points of these reflectors varies at different distances, leading to differences in their imaging size. As the distance between the target and the camera increases, the projection size of the target on the image plane diminishes. Figure 5 illustrates this phenomenon. During calculations, the geometric center of the projection is typically considered as the position of the sphere’s center. Consequently, when there are few imaging pixels capturing the target, errors introduced by the camera are magnified. Some researchers have investigated this error [30]. To mitigate this issue, adjustments to the camera system can be made to ensure that the target is imaged with a specified number of pixels. In other words, the model is abstracted as a segmented function with the distance threshold as the boundary, which is directly related to the physical parameters of the camera. For the OMC system we use, the effective distance is set to 4000 mm.

Apart from camera parameters, another critical factor in triangulation is the view vector angle. The view vector denotes the line that passes through the optical center and connects the image with the object. An intuitive explanation is that when a pair of optical cameras have parallel optical axes and face the same direction, if the two view vectors are nearly parallel with an angle close to 0 degrees, it implies that the target is effectively at infinity. In this scenario, the target exhibits almost no parallax on the imaging planes of the two cameras, making it impossible to calculate depth information. Similarly, if the principal optical axes of the pair of optical cameras coincide and are oriented relative to each other, resulting in an angle of 180 degrees between the view vectors, depth information for the target cannot be derived. This error has been quantified by a group of researchers [27]. If the relationship between imaging *p* and position *P* in space is as follows:(15)$$P=f\left(p\right)$$

Then, the uncertainty of *P* can be expressed as:(16)$$\Lambda P=\frac{\partial f\left(E\left[p\right]\right)}{\partial p}\Lambda p\frac{\partial f{\left(E\left[p\right]\right)}^{T}}{\partial p}$$
where Λp stands for the covariance of *p*.

Their work established an error estimation model for *p* to estimate Λp and calculated the uncertainty in triangulation using the Levenberg–Marquardt algorithm. The relationship between the error and the convergence angle of the view vectors is as follows: when the convergence angle of the view vectors falls within a certain range, the error remains relatively stable and acceptable. However, the error increases rapidly when the convergence angle approaches $0^{\circ }$ and $180^{\circ }$. The commercial binocular optical tracking system we utilize also adheres to this principle, and its operational space is enclosed by a certain amount of planes to ensure that the angles of the target points observed by the camera meet the requirements. Consequently, the model is abstracted as a piecewise function, using the minimum and maximum convergence angles as thresholds. In other words, measurements outside this range are considered to be less credible. In the system we have constructed, this range is set as $40^{\circ }$ to $140^{\circ }$.

### 4.2. Dynamic Evaluation Method Based on View Margin Model

The content proposed in the first section allows for a quick evaluation of the static positioning accuracy of a point in space in a multi-camera system when there is no occlusion. It does not specifically involve calculating the coincidence of view cones or consider the device’s built-in 3-D reconstruction algorithm. However, in real-world scenarios, the true challenge arises from dynamic occlusions that occur during the process. Occlusions caused by the random movement of various objects in the scene cannot be quantified. An intuitive observation is that the measurement accuracy of a target observed by multiple cameras in the center of the scene, with better view vector intersection angles, is higher than that of a target observed by only a few cameras near the threshold angle at the edge of the scene.

The traditional occlusion problem is often framed as a computational geometry challenge, similar to the art museum problem. This problem revolves around determining how to efficiently guard an art museum with the fewest number of guards while ensuring that every corner of the museum remains within the view of these guards. However, when dealing with dynamic scenes and random occlusions, we not only need to consider whether the target can be seen, but also whether it is easy to see. As a result, the concept of view of a margin model is proposed.

Researchers [31] initially introduced the probabilistic occlusion model. They conceptualized occlusion as an infinite plane positioned near the object point and perpendicular to the plane defined by the view vector. This imaginary plane effectively divides the space into two halves, implying that no matter what, the camera on the other side of the occlusion cannot observe the object point. The model then evaluates the likelihood of the object being either visible or blocked when such an occlusion is present. However, it is important to note that the probabilistic occlusion model does not account for other factors that can affect measurements, including the aspects of view vectors, such as angle and resolution degradation, as discussed in Section 1.

For object points and camera pairs in space, assuming that the angle between their view vectors meets the requirements described in Section 1, the intersecting view vectors partition the plane into four regions. When the occlusion rotates within the range of $180^{\circ }-\beta$, triangulation remains unaffected, as illustrated in Figure 6. We refer to this as the view margin of these two cameras.

Building upon the factors mentioned above and the view margin model, we introduce a location evaluation method for assessing the margin of areas in space influenced by occlusion. The pseudo code is presented in Algorithm 1.

**Algorithm 1:** View margin computation for a point in the region of interest

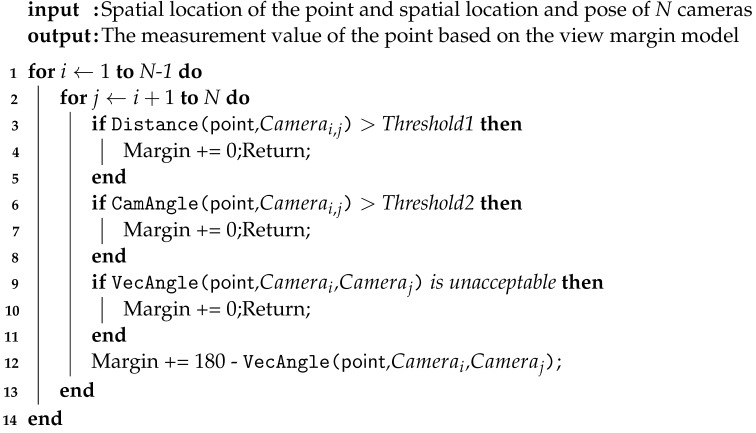



For a point within the region of interest in space, all camera pairs within the camera system are examined. Firstly, the assessment is made to determine whether the Euclidean distance between the camera and the point is impacted by resolution degradation, simplifying the model as a segmented function constrained by a distance threshold. Next, we calculate whether the point falls within the field of view of each camera. Finally, we evaluate whether the view vector between the camera pair and the point falls within an acceptable range, further simplifying the model as a segmented function bounded by two thresholds, signifying that an included angle within a specific range is deemed acceptable. When these static conditions are met, the point is considered visible to the camera pair, and the sum of occlusion rotation margins formed by the point across all visible camera pairs is recorded based on the view margin model.

Regarding local positioning, which operates within the measurement volume of the binocular optical tracking system, we consider its positioning accuracy to be reliable and assign it significant weight. The linkage between the simulation results obtained using the methods described above and the digital twin scene is depicted in Figure 7. It is evident that the areas within the scene can be categorized into three types: the best view volume (BVV), observed by a local positioning device (OTS here); areas not observed by local positioning device but having a good field of vision under global positioning devices (OMC here), leading to a sum of rotation margins surpassing the threshold for the good view volume (GVV); and areas not observed by local positioning device, with poor field of vision under global positioning devices, constituting the poor view volume (PVV) where the sum of rotation margins falls below the threshold.

In certain surgical procedures, such as total knee arthroplasty (TKA), the surgical site may need to be moved extensively, causing the target to exit the measurement range of the local positioning device. Additionally, when the manipulator needs to adjust the end-effector posture or avoid singularities, similar challenges arise. To gather location information for the entire surgical scene, our designed task must fulfill the requirement of collecting multi-target data across the entire workspace. This includes scene targets, such as equipment and medical personnel, as well as important targets like surgical instruments. These important targets must traverse areas with varying accuracy distributions and maintain a relatively stable positioning effect on all targets. In summary, we have established two principles for evaluating positioning methods:Trajectory needs to complete the crossing of BVV, GVV, and PVV areas, and the output data of the positioning method are continuous during the crossing process;The location method should deal with the system deviation of the global positioning itself and the data hop caused by the movement in the PVV during the region switching process.

The determination of the threshold depends on the installation of the camera. In typical usage scenarios, *N* cameras are evenly distributed in the measurement space, with the main optical axis facing the center of the region. Assuming that the cameras are evenly spaced, the angle $\theta^{\prime }$ between the view vectors can be calculated using a trigonometric function, expressed as: (17)$$\theta^{\prime }=\arccos\left(1+\cos\theta \cos^{2}\varphi -\cos^{2}\varphi \right)$$
where φ represents the angle between the view vector (pointing towards the object from the camera) and the ground, θ stands for the angle between the projection of the view vectors on the ground.

If the camera is installed evenly, $\theta \approx 360^{\circ }/N$, the value of φ is between $30^{\circ }$ and $45^{\circ }$. The threshold can be expressed as $k\theta^{\prime }$, where *k* is determined by the settings of the OMC system.

## 5. Experiments

### 5.1. System Setup

The experimental setup is depicted in Figure 8. We installed an optical motion capture system (model Mars 4H from Nokov) in the simulated operating room to provide global information, and an optical tracking system (model Polaris Vega from NDI) was used to provide local information. An industrial robot arm (model Diana from Agile Robots) served as the gold standard for positioning accuracy. At the outset of the experiment, it is ensured that both positioning systems could see the target and the marker representing the global coordinate system provided by us, ensuring proper initialization of the matrix.

We developed a high-speed data processing and synchronization program. Each device used an independent thread to collect timestamped data at specified intervals, synchronized with the system clock of the data processing program. Data were stored in a thread-safe space. After processing and merging data from multiple sources, it was shared with the main process, maintaining synchronization by monitoring real-time processing. This system was developed on a 64-bit Windows 10 OS.

### 5.2. Dynamic Task Evaluation Methods and Results

To validate the effectiveness of the proposed system, we employed the method introduced in Section 4 to preliminarily estimate the view margin in the established digital twin scene. The trajectory of the target was designed to traverse various regions during its motion to analyze the impact of different motion processes on the algorithm. In the table below, we use “P-B-G” to denote the process of the target moving from PVV to BVV, GVV. The threshold value is dependent on the number of cameras and spatial relationships. In this experimental scene, $\theta \approx 50^{\circ }$, k=6, and threshold
$\approx 300^{\circ }$. It is important to note that this value is utilized for the qualitative description of spatial regions and does not involve quantitative calculations of positioning results.

We opted to use a robotic arm to establish ground truth since the accuracy of motion capture systems in complex, occluded environments may not be entirely reliable. The robotic arm utilized in the experiment boasts seven degrees of freedom. By adjusting the position of the robotic arm, we are able to create a trajectory that satisfied the requirements outlined in Table 1 when only a single rotational joint was in motion. During this process, the trajectory of the target affixed to the last joint of the robotic arm resembled a segment of a circular arc, with its accuracy assured by the robot’s encoder, which is highly reliable.

To describe the positioning results, we utilized mean absolute error (MAE). Furthermore, we introduced the compensated term of the Kalman filter (CKF) into the control group.

Table 1 displays the mean absolute error (MAE) results for different tasks employing various methods. From the table, it is evident that our method consistently outperforms the traditional Kalman filter across all scenarios. This superiority can be attributed to the inherent limitations of multi-camera systems in expansive scenes, which struggle to mitigate the inaccuracies caused by irregular occlusions and subsequent distortions post-scene calibration.

When comparing task 2 and task 3, the results for all four methods are quite similar in task 2, as the target ultimately enters the optical tracking system (OTS) measurement volume. However, when transitioning from poor view volume (PVV) to best view volume (BVV), the compensated term plays a significant role. In task 4, where the target enters the good view volume (GVV) after compensation, the disparities among the algorithms are less pronounced under favorable visibility conditions. Nevertheless, upon comparing task 4 with tasks 5 and 6, as visibility deteriorates once more, the compensated term yields better results.

Furthermore, our focus lies on assessing the practical effectiveness of the system in error correction. Within the context of task 1 and the B-P process, we opt to analyze data corresponding to the transition of the target from the best view volume (BVV) to the poor view volume (PVV). The box plot and essential findings are depicted in Figure 9 and elaborated upon in Table 2.

The raw dataset, denoted as OMC, exhibits conspicuous system errors. The traditional filtering method introduces oscillations due to substantial disparities between input values and the current system state during the region transition, with BKF offering a modest reduction in the impact of these oscillations.

Significantly, the incorporation of the compensation term results in a substantial reduction in the system’s maximum error, diminishing from −6.36 mm to −2.02 mm. This adaptation effectively aligns the system’s output with a much closer approximation of the true value, particularly during steady-state operation (as evident from the median).

## 6. Conclusions

In this article, we introduce the innovative concept of a full-perception surgery environment, extending robotic surgery beyond instruments to the entire operating room.

In the construction of the joint positioning framework, optical markers are used as relays for global and local positioning devices, which will introduce some errors. The process of superimposing and calculating measurement errors may result in uncertain systematic errors. We use an improved Kalman filter method to deal with two types of systematic errors: distortion and hops. This method is not a mathematical improvement of the classic Kalman filter; instead, it serves as an application optimization in specific scenarios. The utilization of a partial matrix theoretically enhances the algorithm’s performance. The results in Table 2 demonstrate its responsiveness to significant input changes. Moreover, the method of compensating for the overall result with local positioning has notably improved the system’s output. As for the evaluation method, we employ the view margin model for rapid space classification and design a trajectory that more closely mirrors real-world scenarios—an action trajectory within the visual field space that is not solely constrained by the local OTS. This approach has yielded superior results compared to traditional methods.

Our system assists in mitigating unexpected occlusions: when the optical tracking system’s (OTS) field of view is obstructed, it calculates the compensation amount when it is unobstructed, ensuring that global observation results do not introduce oscillation during iteration. Simultaneously, accuracy is guaranteed (occlusion can be likened to a B-P or B-G process). There is no need to halt the surgical procedure when occlusion occurs, and the occurrence of occlusion can be promptly indicated.

In future research, our aim is to introduce additional data to enhance the overall integrity of the operating room. This includes real-time D-H parameters of the robotic arm and data from other sensors, such as depth cameras, as the specific states of the robotic arm or natural targets (such as humans) cannot be perceived by the infrared optical system. This will further advance the level of human–robot interaction in robotic surgery scenarios. In addition, using evolutionary computing [32] to optimize parameter selection or improve the system’s ability to cope with changing environments based on adaptive learning methods [33] can also help our work.

## Figures and Tables

**Figure 1 sensors-23-08637-f001:**
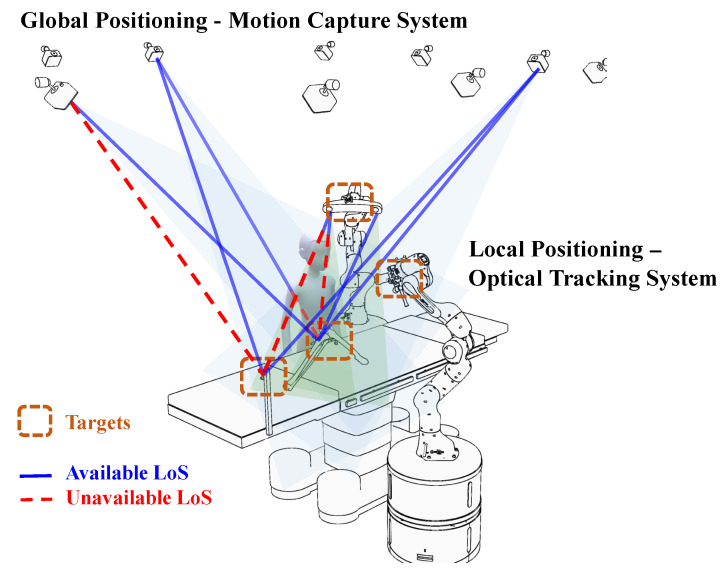
The line-of-sight (LoS) constraints in the robot-assisted surgery scene can be overcome by utilizing global positioning information to assist with local positioning.

**Figure 2 sensors-23-08637-f002:**
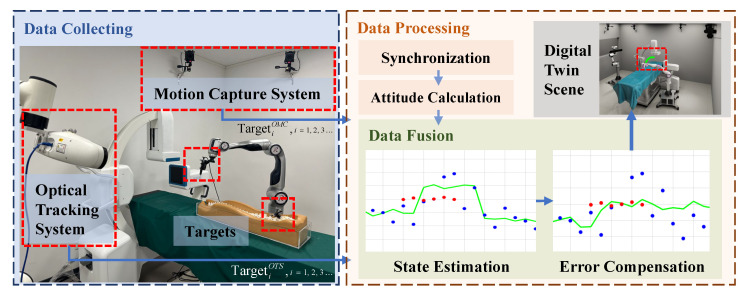
System structure: Global and local positioning systems synchronously collect data, output trajectory information through data fusion algorithms, and drive our digital twin scene.

**Figure 3 sensors-23-08637-f003:**
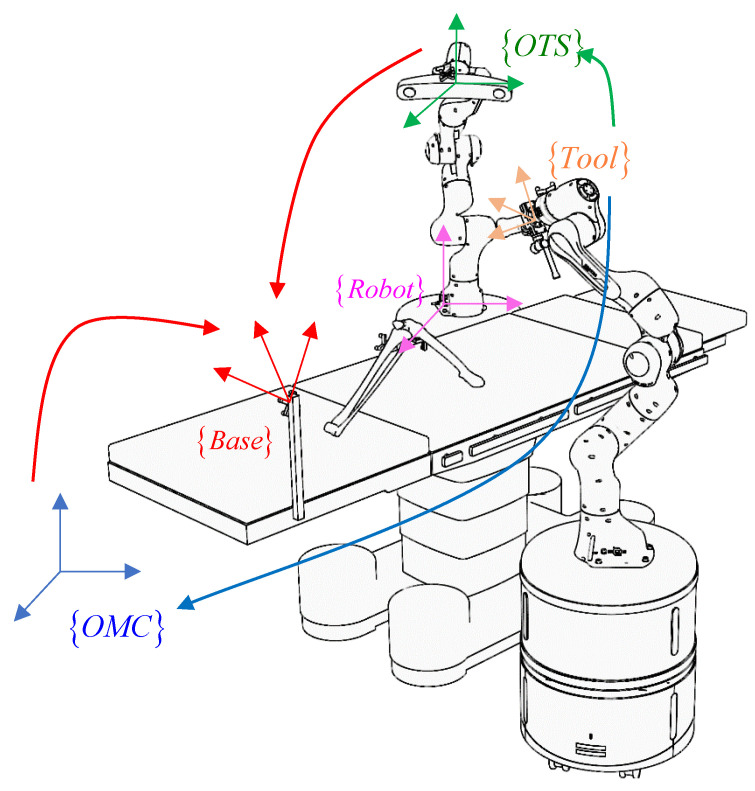
Unify the coordinate expression of targets and devices within the scene through base coordinate markers.

**Figure 4 sensors-23-08637-f004:**
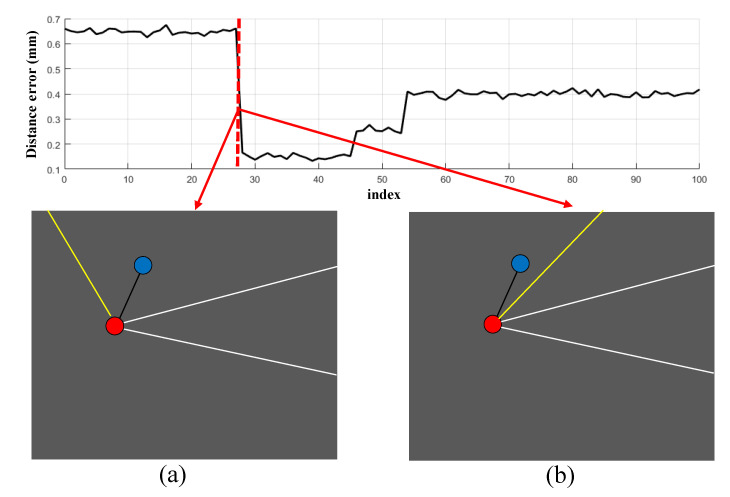
Frames (**a**,**b**) are consecutive in a continuous measurement, and there is a significant discontinuity in the measurement of the distance between the two points. This is attributed to poor visibility conditions.

**Figure 5 sensors-23-08637-f005:**
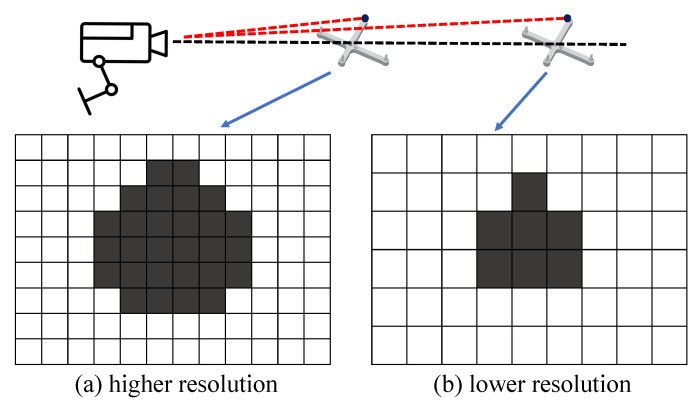
The calculation of the geometric center of a retro-reflective sphere by an optical camera is influenced by the distance.

**Figure 6 sensors-23-08637-f006:**
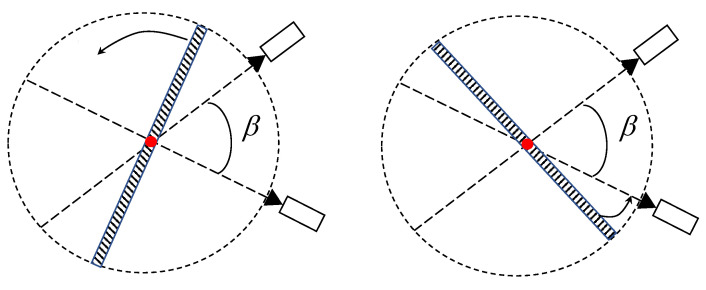
When the camera angle is β, the tolerance for occlusion that does not affect the field of view is $180^{\circ }-\beta$.

**Figure 7 sensors-23-08637-f007:**
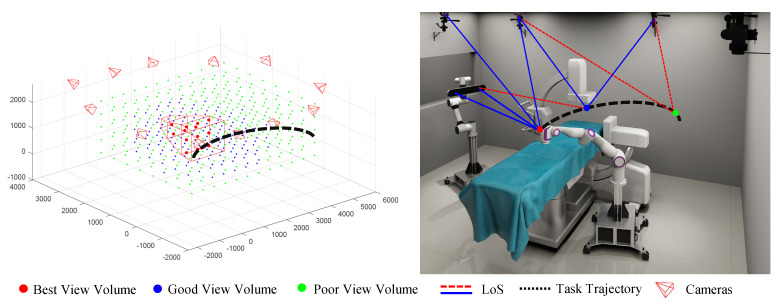
View margin distribution and digital twin scene generated based on view margin model.

**Figure 8 sensors-23-08637-f008:**
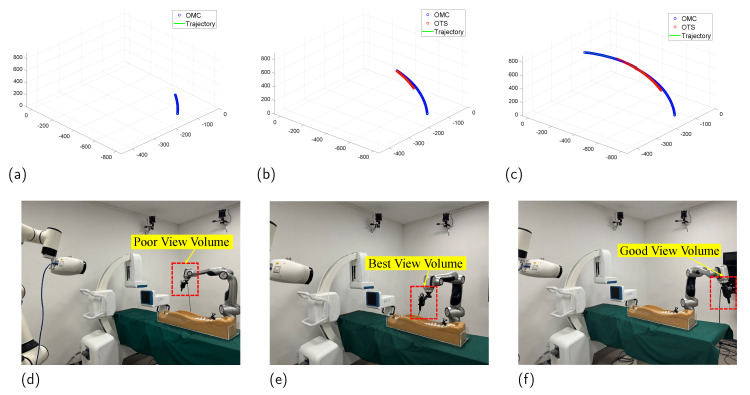
Data collection and actual experimental scenario for Task 4: The robotic arm grasps the target, moves from the PVV to the measurement volume of the OTS (BVV), and then moves out of BVV to reach the GVV; (**a**–**c**) show the collected data and fused results, and (**d**–**f**) demonstrate the actual scene of this process. We can see distortion between OTS and OMC measurement data.

**Figure 9 sensors-23-08637-f009:**
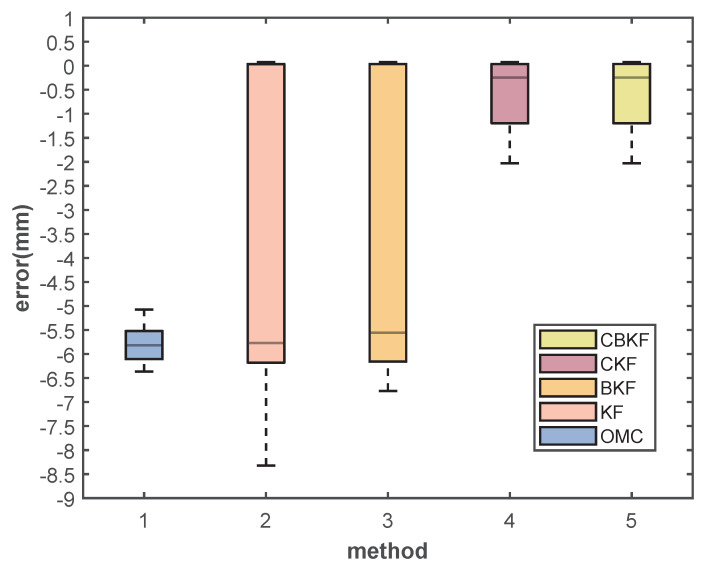
Box plot of errors in entering PVV from BVV.

**Table 1 sensors-23-08637-t001:** MAE of different methods (mm).

ID	Task	OMC	KF	BKF	CKF	CBKF (Ours)
1	G-B-P	3.39	1.84	1.82	1.64	**1.63**
2	P-G-B	3.00	1.99	1.98	1.99	**1.98**
3	P-G-B-P	3.94	2.61	2.59	1.85	**1.85**
4	P-B-G	3.79	2.02	2.02	2.02	**2.02**
5	B-G-P	3.79	2.14	2.13	1.82	**1.82**
6	P-B-G-P	3.81	2.21	2.20	1.77	**1.76**

**Table 2 sensors-23-08637-t002:** Comparison of values from different methods (mm).

Value	OMC	KF	BKF	CKF	CBKF (Ours)
max	6.36	8.32	6.76	2.02	**2.02**
median	5.81	5.77	5.56	0.25	**0.25**

## Data Availability

Data sharing is not applicable to this article as no new data were created or analyzed in this study.

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
