# Peer review of "Full-Perception Robotic Surgery Environment with Anti-Occlusion Global–Local Joint Positioning"

_sensors, 2023, doi:10.3390/s23208637_

Round 1
Reviewer 1 Report
The authors proposed the concept of full-perception robotic surgery and studied the combination of optical motion capturer and optical tracking system in producing a digital twin for surgical robots. They built a system to link the local and global coordinate systems, introduced biased Kalman filter to deal with occasional jumps of the tracked markers and proposed a method to evaluate the tracking quality in the operation room. Overall, this work is well organized and presented. I think it could be accepted after a minor revision. Here are my detailed suggestions:
- It is better to explain the dots and lines in the caption of Fig.4. It’s hard to understand the data solely from the subfigures.
- I suggest to add a figure to help explain Eqn.17
- It is difficult to understand Eqn.16 without reading the reference [30]. To make this work self-explained, I suggest to better introduce the terms in Eqn.16.
The English writing is good.
Author Response
Dear Reviewer,
Thank you for your valuable review comments. Our response letter is attached.

Reviewer 2 Report
Authors presented a global-local joint positioning method based on the data fusion of a motion capture system and a binocular optical tracking system. The work presented in this paper is well organized, however a few recommendations are required to be addressed:
1. I suggest adding more results, in addition, authors need to present a set of validation metrics.
2. Authors need to compare their results with the results obtained from the recent developed relevant systems.
3. Authors need carefully to read their paper. For instance, line 359 includes: "...... in Chapter 4 preliminarily ....."
The presented work is well written, however authors need to proof-read their work.
Author Response

(The authors gave the same response as above.)

Reviewer 3 Report
In the robotic surgery environment, individuals, robots, and medical devices move relative to each other, which leads to unforeseen mutual occlusion. The absence of positioning information can impact behavioral decision-making and coordination. On the whole, the research content of this manuscript is relatively meaningful, but there are also some problems that need to be modified.
1. To objectively explain the content of the manuscript from the perspective of a third party, it is recommended to avoid using ‘we’ in this manuscript as much as possible. Meanwhile, it is also recommended to change the sentences related to ‘we’ to the passive voice of the present tense.
2. To highlight the new methods proposed in this manuscript, as well as the new achievements achieved, the reviewer suggests reorganizing the abstract. At present, the content of abstract appears relatively flat.
3. The conclusions of this manuscript in Section 6 are relatively simple and do not provide a good overview of the results achieved. It is recommended to further organize and enrich the content of this section.
4. To effectively highlight the superiority of the methods proposed in this manuscript, it is recommended to further extend the content of Table 1. The reviewer suggests using a comparative approach to present the superior or comparative values of the methods proposed in this manuscript, such as A being X% higher than B, which is more intuitive.
5. To facilitate readers' better understanding for the research content of this manuscript, it is recommended to rewrite or reproduce the section structure of this manuscript in the last paragraph of Section 2.
To objectively explain the content of the manuscript from the perspective of a third party, it is recommended to avoid using ‘we’ in this manuscript as much as possible. Meanwhile, it is also recommended to change the sentences related to ‘we’ to the passive voice of the present tense.
Author Response

(The authors gave the same response as above.)

Reviewer 4 Report
The authors have presented an absolutely novel, relevant and well-written manuscript. The Introfuction is concise and presents the development of the work described below.
The mathematical formulation is concise and well written.
Author Response
Dear Reviewer,
Thank you for your valuable review comments.
Reviewer 5 Report
-
- 1. Technical Detail: The manuscript presents a global-local joint positioning method for a full-perception robotic surgery environment. However, there seems to be a lack of clarity in the technical details surrounding the established framework. Elucidating on how the motion capture system and binocular optical tracking system synergize within this framework could provide readers with a clearer understanding.
- 2. Comparative Analysis: The paper briefly mentions that the presented method yields better positioning results than classical filtering methods. A deeper comparison with "EGNN: Graph structure learning based on evolutionary computation" might offer insights into how evolutionary computation could be incorporated into the fusion method for enhanced performance.
- 3. Interdisciplinary Integration: The topic of robotic surgery inevitably overlaps with various advanced algorithms and methods. Insights from the "Output-Feedback Robust Tracking Control of Uncertain Systems via Adaptive Learning" might suggest how adaptive learning could potentially enhance the positioning accuracy in dynamic occlusion environments.
- 4. Network Representation & Blockchain Technology: Considering the increasing prominence of blockchain in various sectors, integrating ideas from "Heterogeneous Network Representation Learning Approach for Ethereum Identity Identification" could pave the way for ensuring the secure transmission and storage of data within the robotic surgery environment.
- 5. Practical Implications: While the paper touches on the innovative concept of a full-perception surgery environment, it would be beneficial to delve deeper into its practical implications. For instance, how does the global-local joint positioning approach cater to sudden, unexpected changes in the surgical environment?
- 6. Model Evaluation: The manuscript mentions a method to evaluate positioning accuracy based on the view margin model. An in-depth explanation of how this model differs from or improves upon existing evaluation models would add depth to the narrative.
- 7. Future Directions: The conclusion section hints at future work involving the integration of data from robotic arm control and other agents. It would be insightful to provide preliminary thoughts or directions on how these data sources could be effectively amalgamated into the existing framework.
Author Response

(The authors gave the same response as above.)

Round 2
Reviewer 2 Report
The authors have addressed the raised issues. However, a table number is missing in the Conclusion part.
"The results in Table ?? demonstrate its responsiveness to significant input changes"
NA
Author Response
Dear Reviewer,
I wish to express our sincere gratitude for your meticulous review of our manuscript. We would like to extend our sincerest apologies for the errors found in the manuscript. The oversight occurred when we were drafting our response letter within the same document and inadvertently copied the table number from the response letter when submitting the final version of the manuscript. Regrettably, this led to a reference to a non-existent table, which was a result of our failure to thoroughly inspect the manuscript. Rest assured, we have now rectified this mistake.
Once again, we appreciate your thorough examination of our work and thank you for your valuable feedback.
Yours sincerely,
Reviewer 5 Report
The authors have addressed all of my concerns. The current version can be accepted.
Author Response
Dear Reviewer,
Thank you for your careful review and suggestions.
Yours sincerely,